**Data Availability Statement:** The gold price dataset is available at https://github.com/amirhosseinse7/Closing-gold-price.

**Funding:** The authors received no specific funding for this work.

# Gold price prediction by a CNN-Bi-LSTM model along with automatic parameter tuning

**Amirhossein Amini[1], Robab Kalantari** [2]*

1 Faculty of Industrial Engineering, Department of Financial System, Khatam University, Tehran, Iran,
2 Faculty of Finance, Department of Financial Engineering, Khatam University, Tehran, Iran

* r.kalantari@khatam.ac.ir

## Abstract

Banking and stock markets consider gold to be an important component of their economic and financial status. There are various factors that influence the gold price trend and its fluctuations. Accurate and reliable prediction of the gold price is an essential part of financial and portfolio management. Moreover, it could provide insights about potential buy and sell points in order to prevent financial damages and reduce the risk of investment. In this paper, different architectures of deep neural network (DNN) have been proposed based on long short-term memory (LSTM) and convolutional-based neural networks (CNN) as a hybrid model, along with automatic parameter tuning to increase the accuracy, coefficient of determination, of the forecasting results. An illustrative dataset from the closing gold prices for 44 years, from 1978 to 2021, is provided to demonstrate the effectiveness and feasibility of this method. The grid search technique finds the optimal set of DNNs' parameters. Furthermore, to assess the efficiency of DNN models, three statistical indices of RMSE, RMAE, and coefficient of determination ($R^2$), were calculated for the test set. Results indicate that the proposed hybrid model (CNN-Bi-LSTM) outperforms other models in total bias, capturing extreme values and obtaining promising results. In this model, CNN is used to extract features of input dataset. Furthermore, Bi-LSTM uses CNN's outputs to predict the daily closing gold price.

## 1. Introduction

Gold is one of the most critical minerals in economics and politics because central banks of countries hold gold reserves as a guarantee to pay for trade on the world market. Moreover, gold is the most popular choice for investments among all commodities [1]. However, there are significant volatility and strong gold price fluctuations due to various reasons such as supply and demand, political issues, and inflation. Developing an accurate closing gold price prediction is not a simple task due to its nonlinear nature, the large number of unpredictable factors involved, and the significant volatilities of the gold price, resulting in complicated temporal time series [2]. Literature reviews show that several traditional methods have been developed for gold price forecasting, such as linear regression, support vector regression (SVR), autoregressive moving averages (ARMA), autoregressive integrated moving averages

**Competing interests:** The authors have declared that no competing interests exist.

(ARIMA), and artificial neural networks (ANNs) [3–6]. However, statistical models are usually based on stationarity and linear correlation assumptions. In addition, machine learning models appear to be unable to detect and capture the nonlinear and complex behavior of the gold price time sequence. Therefore, none of these methods is able to provide a robust and reliable forecasting model [7].

DNNs address ANNs' shortcomings, such as gradient vanishing. In addition, the automatic learning capabilities of DNNs enable them to find relationships between input and output variables of raw data on a complex and high-level basis [8]. In recent years, the development of DNNs for the prediction of the gold price has gained popularity among the scientific community [9]. There has also been widespread use of CNNs and recurrent neural networks (RNNs) to model time series [10]. Using these models, it is possible to enhance the accuracy of time sequence prediction by addressing the problem of long-term dependencies in the input data using the LSTM architecture design and the CNN models by eliminating the noise from the input data and extracting more useful features for the prediction [11]. Although many studies have been conducted on gold price forecasting using DNNs, the automatic parameter tuning (grid search method) of DNNs considering sensitivity analysis for closing gold price has not been collectively studied. The main objective of this paper is to propose a tuned hybrid deep neural network for the adaptive prediction of closing gold prices. Specifically, the model predicts the gold price for the next time step based on the analysis of historical gold prices. The proposed model was assessed against state-of-the-art DNNs such as CNN, CNN-LSTM, Conv-LSTM, and Stacked LSTM for the adaptive prediction of closing gold price. Using the one parameter at a time method for sensitivity analysis, we also analyze the effect of look back, learning rate, and mini-batch size parameters.

The rest of this paper is organized as follows: "section 2: Related work" summarizes recent studies on the application of various models and parameter tuning for gold, stock, and Bitcoin price prediction. "Section 3: Methods" presents a general description of DNNs, grid search technique, statistical criteria, and the dataset utilized in the study. "Section 4: Proposed CNN-Bi-LSTM model" provides a detailed description of the proposed deep learning model. In "Section 5: Results and discussion", the modeling results and sensitivity analysis are presented. "Section 6: Conclusion" summarizes the research findings, and some outlines are provided for future research.

## 2. Related work

The scientific community has grown increasingly interested in using DNNs and other models to predict gold's price movement over the last decade. The introduced methodologies provided some useful insight into the closing gold price behavior. Pindoriya et al. [12] used an adaptive wavelet neural network (AWNN) to predict energy price in electricity markets. Results showed that AWNN predicted the sequence more accurately than other methods such as wavelet-ARIMA, multi-layer perceptron (MLP), and radial basis function (RBF). Vidya & Hari [13] predicted gold price time series, 10850 data from 1979 to 2020, using LSTM and CNN deep neural networks. They utilized an adaptive moment estimation (Adam) optimizer, which could efficiently deal with nonlinear problems such as gold price prediction. Researchers reported that LSTM forecasts were superior to common models, such as ARIMA, SVR, and CNN. Yadav et al. [14] predicted the real-time stock price using the best features of Fast Recurrent Neural Networks (FastRNN), CNN, and Bi-LSTM algorithms. The model predicted the stock price for daily time steps using four companies' minute time step data. The proposed model outperformed ARIMA and other real-time hybrid DNNs. Li et al. [15] proposed an LSTM model to forecast the daily stock price in China using limit order books (LOBs) data.

Based on experimental results, they highlighted the effectiveness of the proposed model. Madziwa et al. [16] predicted the annual gold price using autoregressive distribution lag (ARDL), stochastic mean reverting, and ARIMA models. Results demonstrated that ARDL model performed better compared to other models.

An important point to note is that many of the datasets used in these studies include recent COVID-19 pandemic values at the beginning of 2020, accompanied by substantial volatility and departures from usual behavior. Mohtasham Khani et al. [17] conducted a gold price prediction on daily time steps during the Coronavirus pandemic using DNNs, namely LSTM, CNN, Bi-LSTM, and CNN-LSTM models. They tried to explore the effect of new daily cases of Coronavirus on financial markets, especially gold price prediction using data collected on the gold price and new daily cases of Coronavirus from 2015 to 2020. Based on the mean squared error (MSE) indicator, the authors' highlighted the potential of Stacked LSTM compared to CNN-LSTM and Bi-LSTM for adaptive and multi-step gold price forecasting. Hansun & Suryadibrata [18] utilized a simple three-layer LSTM to predict gold price during the COVID-19 crisis using 5000 data. The LSTM parameters, such as the number of hidden units, batch size, and dropout probability are selected using the trial and error approach. They reported that the determination coefficient and RMSE of the LSTM network equal 97.2% and 39.94, respectively.

According to studies, hybrid DNNs are more effective than individual DNNs. For example, He et al. [19] predicted daily gold price based on the integration of the LSTM and CNN models. The results indicated that their model outperforms traditional models such as SVR, ARIMA, CNN, and deep regression. Li & Dai [20] proposed a CNN-LSTM network for predicting closing Bitcoin prices based on the closing Bitcoin price itself, macroeconomic indicators, and investors' data. The results illustrate that the CNN-LSTM hybrid network can effectively enhance value and direction prediction accuracy compared to backpropagation (BP), CNN, and LSTM. Livieris et al. [9] used a multivariable CNN-LSTM model for the prediction of cryptocurrencies such as Bitcoin (BTC), Ethereum (ETH), and Ripple (XPR) and their movements. The results showed that the proposed hybrid DNN has the ability to efficiently reduce overfitting and computational cost compared to traditional fully connected DNNs. Lu et al. [21] proposed a hybrid DNN (CNN-Bi-LSTM) to predict the daily closing stock price. The optimal parameters and hyperparameter of the CNN-Bi-LSTM model are hand-picked through a trial and error approach. They choose the Adam optimizer with a 0.001 learning rate and loss function equal to MAE. Results demonstrate that the performance of this model is superior to the performance of MLP, LSTM, CNN, Bi-LSTM, and CNN-LSTM with MAE, RMSE, and $R^2$ are 21.95, 31.69, and 0.98 respectively. Kim and Jang [20] used hybrid deep neural networks (CNN-LSTM and CNN-GRU) to predict the price of crude oil. Results show that their proposed models are more accurate (0.985 and 0.988 correlation coefficients) than other DNNs. Zhao and Yang [22] proposed a new hybrid LSTM based model to predict stock price movement prediction. Evaluations demonstrated that the accuracy of proposed model is higher compared to other models. Chen et al. [23] used the combination of the K-means clustering algorithm and LSTM to predict China's commercial bank stock price. A K-means method was utilized to find other banks with similar price trends, which were then used as input sequences to an LSTM network in order to predict stock prices over the next multiple time intervals. The researchers discovered that their methodology exhibited a superior level of accuracy, as evidenced by lower MSE and MAE values and higher $R^2$, compared to alternative approaches in the prediction of stock prices. Liu et al. [24] utilized a compound of variational mode decomposition and metal-learning algorithms to predict stock price. First, they broke the sequence into smaller sequences using the variational mode decomposition method, then the They first broke down the sequence to subsequences using variational mode

decomposition, then used the model-agnostic meta-learning algorithm and LSTM network to forecast the subsequences. Finally, the subsequences predictions are combined to produce the final output sequence. According to the results, this method performed better (higher coefficient of determination and lower RMSE) than other methods for different datasets. Vo et al. [11] predicted the Brent oil price using the Bi-LSTM network. Researchers reported that Bi-LSTM Network outperformed other networks such as LSTM, CNN-LSTM, and CNN-Bi-LSTM. Lu et al. [25] predicted the gold price using a hybrid model called complete ensemble empirical mode decomposition with adaptive noise (CEEMDAN)- grey wolf optimizer (GWO)- SVR and high-frequency intrinsic mode functions subsequence (IMFH), low-frequency intrinsic mode functions subsequence (IMFL), and the residual as input data. The IMFL and the residual have higher correlations compared to IMFH. These factors showed the prominent factors and overall trend of gold price sequence. The final output is gained through combining the aforementioned timeseries. Results showed that the proposed model outperforms other models such as ARIMA, generalized auto-regressive conditional heteroscedasticity (GARCH), ANN, classification and regression tree (CART), and BPNN (back propagation neural network). Zhao and Yang [26] employed emotion-enhanced CNN, denoising autoencoder (DAE), and LSTM models to forecast stock prices. The findings indicate that the combined model exhibited superior accuracy compared to the LSTM and CNN base models. Madhika et al. [27] applied the ARIMA and LSTM models for the purpose of forecasting gold prices. The findings indicate that the LSTM model outperforms the ARIMA model in terms of the RMSE and Mean Absolute Percent Error (MAPE) statistical indicators. Pangestu et al. [28] employed a Long Short-Term Memory (LSTM) model and a Fuzzy system in their study to forecast the price of gold. The results of the study revealed that the RMSE of the LSTM model exhibited superior performance compared to the fuzzy system. Liang et al. [29] used the improved version of CEEMDAN (ICEEMDAN) algorithm at first stage. Next, they forecasted the gold price using the long short-term memory, convolutional neural networks, and convolutional block attention module (LSTM-CNN-CBAM) for different sublayers. They reported the proposed model (i.e., ICEEMDAN-LSTM-CNN-CBAM (ILCC)) exhibited a superior level of accuracy (R-square) compared to other related models. The use of individual and hybrid DNNs for gold price prediction has been extensively studied; however, the comparison between various DNNs, the use of CNN-Bi-LSTM architecture for closing gold price prediction, the automatic parameter tuning using grid search to determine optimal parameters of each DNN, and the sensitivity analysis of three hyperparameters of the best DNN have not been appropriately explored. Our research contribution addresses the aforementioned shortcoming in closing gold price prediction.

## 3. Methods

In this section, four deep learning networks, grid search hyperparameter tuning technique, statistical evaluation, and data used for closing gold price forecasting are explained in detail.

### 3.1 Deep learning models

**3.1.1 Stacked LSTM.**   The LSTM network, an artificial RNN, is first introduced by S. Hochreiter and J. Schmidhuber in 1997 [30]. This network is designed to address the problem of long-term dependencies in time sequences and gradient explosion and disappearance in RNNs [31]. This architecture has loops so it can retain information or omit unneeded information. The network adjusts information flow by using gate control mechanisms and systematically determines how much incoming information is retained in each step [32]. The stacked LSTM consists of multiple LSTM layers with specified hidden units stacked on top of each

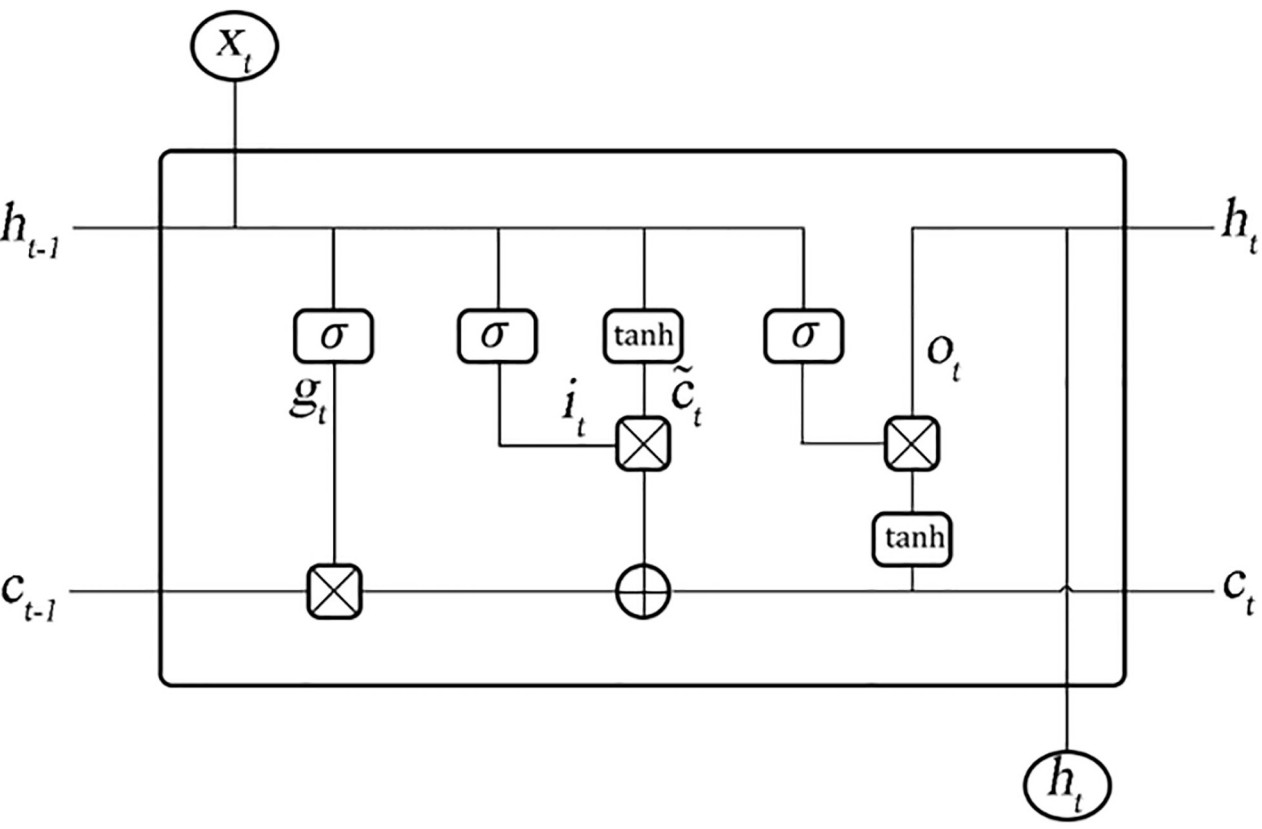

**Fig 1. The schematic figure of the Long-Short Term Memory (LSTM) network.**

other. Fig 1 illustrates the three parts of the LSTM memory cell: the input gate, the forget gate, and the output gate. Input gates ($i_t$) control the flow of new values into the cell and store some information about the current value. The forget gate ($g_t$) determines whether values need to be retained or forgotten. Also, the output gate ($o_t$) select useful information from the current cell state. The outputs of each cell that being transferred to the next cell are hidden state ($h_t$) and cell state ($c_t$) [13].

U and W are weight matrices in the below equations, b is a bias term, ($\sigma$) is a sigmoid activation function, and the symbol * represents multiplication. The following equations present the architecture of the LSTM network.

$$i_t = \sigma \left( U_i x_t + W_i h_{t-1} + b_i \right) \tag{1}$$

$$g_t = \sigma \left( U_g x_t + W_g h_{t-1} + b_f \right) \tag{2}$$

$$o_t = \sigma \left( U_o x_t + W_o h_{t-1} + b_o \right) \tag{3}$$

Eqs (1)–(3) represent input, forget, and output gate expressions, which take the weighted sum of the hidden state at time t-1 ($h_{t-1}$), and the input at time t ($x_t$) as inputs of activation functions to determine the output [33]. The Eqs (4)–(6) are as follows for the candidate state ($\tilde{c}_t$),

cell state ($c_t$), and final output or hidden state at time t ($h_t$).

$$\tilde{c}_t = \tanh\left(U_c x_t + W_c h_{t-1} + b_c\right) \tag{4}$$

$$c_t = g_{t^*} c_{t-1} + i_{t^*} \tilde{c}_t \tag{5}$$

$$h_t = o_{t^*} \tanh\left(c_t\right) \tag{6}$$

**3.1.2 CNN.** The convolutional neural network (CNN) consists of many layers, each with its distinct architecture, called convolutional layers and pooling layers [34]. It emulates the perception of local information using local information by biological vision cells. The data features are extracted using a local connection and layer-by-layer computation, and then the global information is synthesized with the full connection [20]. In convolutional-based networks such as ConvLSTM and CNN-LSTM, CNN is used to extract features of input time series efficiently and learn the internal representation of sequence [35]. Several parameters are required for CNN architectures, such as the stride type, pooling type, and filter size. The filter size shows the set of inputs to which the kernel will be applied during the convolution process. The stride is the number of steps the filtering process will take. The pooling type indicates which pooling process will be applied to the filtered output [36]. Convolution layers contain a plurality of convolution kernels, and their calculation is shown in Eq (7). The extracted features of the convolutional layer are very high. After the convolution layer, this problem is solved by adding a pooling layer to reduce the feature dimensions and the cost of training the network [37].

$$l_t = \tanh(x_t * k_t + b_t) \tag{7}$$

In Eq (7), $l_t$ is the output of the convolutional layer, $x_t$ is the input vector, $b_t$ is the bias of the convolution kernel, tangent hyperbolic (tanh) as the activation function, and $k_t$ is the weight of the convolution kernel.

**3.1.3 CNN-LSTM.** In many applications, such as image recognition, signal processing and natural language processing, convolutional neural networks (CNNs) showed good results. In short-term forecasting time sequences, LSTMs performed well. As a result of combined CNN and LSTM layers, the CNN-LSTM method can extract complex features from datasets and identify complex trends. CNN layers consist of an input layer that receives input time series, an output layer that extracts features for LSTMs, and a certain number of hidden layers. There is usually a convolutional layer, ReLU activation function, and a max pooling layer in the hidden layer. In the CNN layer, local features are extracted from high layer inputs and transferred to the lower layers in order to produce more complex features [38].

**3.1.4 ConvLSTM.** ConvLSTM has been introduced for forecasting temporal sequences, where both input and target are temporal sequences [39]. Additionally, it can capture temporal patterns of large-scale sequential datasets [40]. A convolution operation is applied between raw input data and convolution kernels to produce new features [7]. In addition to its accomplishments in image and speech detection, speech synthesis, and facial recognition, this network has demonstrated outstanding capabilities in exploring time series data [36].

The learning rate is the most important parameter to configure in the tuning process of DNNs. Large learning rates reduce the model's run time at the cost of converging a sub-optimal solution. However, the small learning rate may converge to the global optimum solution of the model with a significantly longer run time. The grid search approach can help highlight the optimal learning rate and describe the effect of the learning rate on the model's

performance [41]. By combining Adaptive Gradient (AdaGrad) and Root Mean Square Propagation (RMSProp), adaptive moment estimation (Adam) calculates adaptive learning rates for each parameter. Adam's algorithm maintains an exponentially decaying average of past gradients, similar to momentum, in addition to an exponentially decaying average of past squared gradients, like Adadelta and RMSprop. In practice, Adam works well and compares favorably to other stochastic optimization methods [42]. In all layers except the output layer, where linear activation was used, the Rectified Linear Unit (ReLU) activation function was employed.

Networks combined with dropout layers are less prone to overfitting by randomly removing hidden units from a network with the tuned probability. As a result, useful features for producing correct answers operated [43]. The number of "lookback" is another important hyperparameter of DNNs. Each combination of parameters has a different effect on the performance of the gold price prediction model. Though using conventional hyperparameters and parameters generate good forecasts in DNNs, we apply automatic parameter tuning to find the optimal performance of these models.

## 3.2 Hyperparameter tuning using the grid search

Many hyperparameters are associated with DNNs, including mini-batch size, learning rate, lookback, etc. Parameters that cannot be updated during the model's training are known as hyperparameters [44]. This can result in suboptimal results if the hyperparameters and parameters are selected incorrectly. Hyperparameter tuning may also be referred to as the process of choosing the most appropriate values for hyperparameters and parameters. Generally, parameters were selected as rules of thumb in the literature of forecasting papers [45].

For deep learning hyperparameter and parameter tuning, a vast potential space must be explored [46]. For example, the parameters of the ConvLSTM network are the number of hidden units, the filter size, mini-batch size, learning rate, and the dropout probability. In general, applying parameters using the rule of thumb or previous research results in the best results, but we have tried to find the best parameter to achieve optimal performance, using automatic parameter tuning for DNNs [47]. There are different automatic hyperparameter tuning algorithms including search, random search, and Bayesian optimization. In this paper, the grid search algorithm is used to adjust the parameters of the DNN. The grid search explores the best combination of parameters within the specified number of trials.

Grid search is one of the basic methods for parameter optimization. Grid search performs an exhaustive search on the set of parameters specified by the user. The users generate all candidates, so they must have preliminary knowledge about these parameters. Grid search may be applied to several parameters with a limited search space. Grid search is the most straightforward algorithm that will always yield the best results as long as sufficient resources are available [44]. In comparison to other tuning techniques, grid search runs in parallel because each trial runs separately without affecting the time sequence. In other words, results for one trial are not affected by those from other trials.

The grid search approach is usually preferred when users are familiar with these parameters enough to define a narrow search space. Despite the fact that other search algorithms may offer more favorable features, grid search remains the most popular method due to its mathematical simplicity [48].

In this study, the maximum number of trials is set at 50. The following parameters, based on previous studies [18,22], ranges are used:

- The number of hidden units per layer related to LSTM and Dense layers is assigned to an integer ranging from 10 to 50 with 10 steps

**Table 1. Formulation of the selected statistical criteria.**

| Statistical criteria | Formula | Description | Perfect/no skill |
|---|---|---|---|
| Coefficient of determination ($R^2$) | $1 - \dfrac{\sum_{i=1}^{N}\left(G_{f_i} - G_{o_i}\right)^2}{\sum_{i=1}^{N}\left(G_{f_i} - \bar{G_f}\right)^2}$ | Agreement on forecasted and observed gold prices | 1/-Inf |
| RMAE | $\sqrt{\frac{1}{N}\sum_{i=1}^{N}\left(G_{f_i} - G_{o_i}\right)}$ | The magnitude of the difference between forecasted and observed gold prices | 0/ |
| RMSE | $\sqrt{\frac{1}{N}\sum_{i=1}^{N}\left(G_{f_i} - G_{o_i}\right)^2}$ | The closeness between forecasted and observed gold prices | 0/ |

Note: $G_{f_i}$ and $G_{o_i}$ are the values of the forecasted and observed dataset for the ith of N closing gold price. Also, $\bar{G_f}$ is the mean of the observed closed gold price dataset.

- The mini-batch size is set to the power of two numbers from 8 to 1028

- The learning rate of the Adam optimizer is set to choose from one of four choices [0.05, 0.005, 0.0005, 0.00005]

- The lookback is assigned to a value ranging from 1 to 40 with one step

- The filter size in convolutional-based networks is set to an integer ranging from 4 to 128 with 12 steps

- The kernel size in convolutional-based networks is set to choose from one of four choices [1, 2, 3, 4]

- The dropout probability is assigned to an integer ranging from 0.1 to 0.9 with 0.1 steps

### 3.3. Statistical evaluation

Three statistical criteria, namely Root Mean Square Error, Root Mean Absolute Error (RMAE), and coefficient of determination ($R^2$), are utilized to evaluate the performance of DNNs (Table 1).

Because deep learning networks are stochastic, five realizations were done for each model, and the results were presented as box plots in the result and discussion section.

### 3.4 Data

Deep neural networks (DNNs) are used to predict closing gold prices based on reference closing gold price data. From 1978 to 2021, the closing gold price from the last 44 years, including the Covid-19 era, is used as an input into models. In standard ANNs, time series are entered into the network in a different way than in RNNs. Standard ANNs are typically trained with random input data. The data in time-dependent networks such as DNNs, however, is presented in occurrence time order to maintain the time-dependency structure. Out-of-sample prediction is the standard method for testing time-dependent networks. Extrapolation beyond the reference dataset constitutes true out-of-sample prediction [49]. For evaluation purposes, the closing gold price data were divided into training, validation, and testing sets (Fig 2). More analytically, the first 70% of the acquired daily data, from 29 December 1978 to 11 September 2008, is used for model training. The following 20% and 10% of the gold price data are selected for model validation (from 12 September 2008 to 8 March 2017) and testing (from 9 March 2017 to 4 June 2021), which ensured a substantial amount of out-of-sample data that had previously not been seen by models.

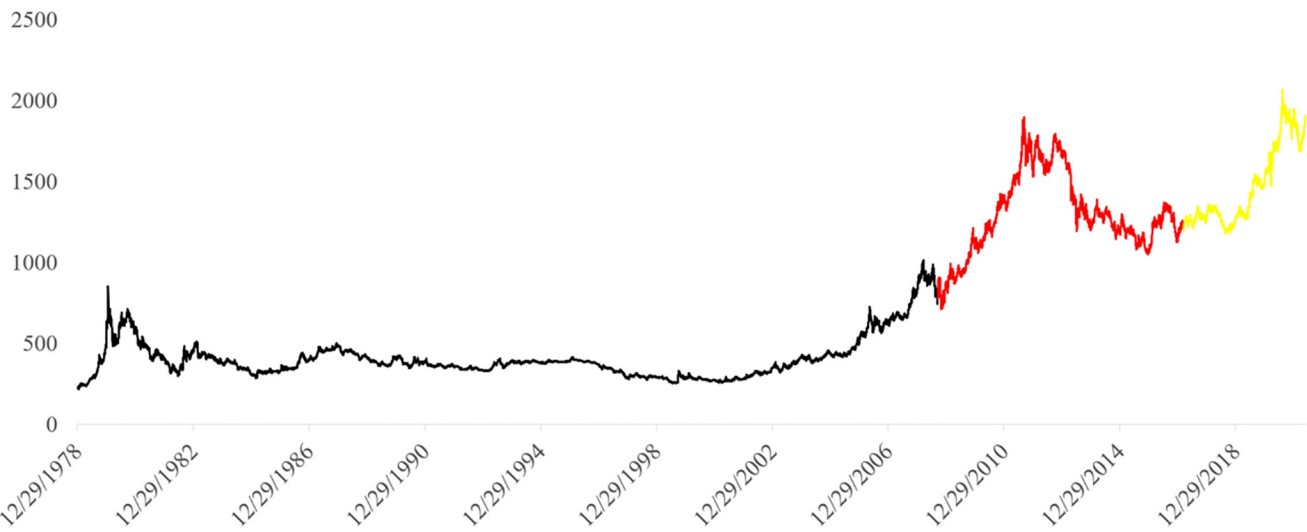

**Fig 2. Daily closing gold price in USD from December 1978 to 4 June 2021 (The black, red, and yellow lines show the training, validation, and test data).**

## 4. Proposed CNN-Bi-LSTM model

By using CNN-Bi-LSTM as a combined model, the CNN and Bi-LSTM models can extract valuable information with the convolutional layer and learn from forward as well as backward time sequences by utilizing LSTM [32]. Using its architecture consisting of forward and backward LSTM layers, Bi-LSTM processes data in both directions—from first to last input and last to the first output and uses both forward and backward information [50,51].

Fig 3 illustrates the flowchart of the proposed methodology for predicting the closing gold price using DNNs and the grid search tuning technique. According to Fig 3, our method is comprised of five steps. The first step is to acquire the daily closing gold price. The reference dataset is then normalized to minimize sequencing depth differences, improve the initialization of parameters, and speed up the convergence of the model. Next, grid search algorithms are used to tune the parameters of adaptive DNN models. Three statistical indices are used to evaluate the performance of the models in the evaluation phase. Finally, the best adaptive model is chosen to be implemented to forecast the closing gold price.

## 5. Results and discussion

### 5.1 Model optimization

The first step is to select the DNN architecture that is best suited to the problem at hand [52]. We will apply the grid search technique in this study to optimize the parameters of each DNN, resulting in optimal performance of the models. The parameters' optimal values are shown in Table 2. According to Fig 4, the CNN-Bi-LSTM network outperforms all other models in the comprehensive comparison between five different state-of-the-art forecasting approaches (Stacked-LSTM, CNN, CNN-LSTM, ConvLSTM, and CNN-Bi-LSTM). The general rule of thumb for manually tuning a model's structures is to start with a simple model, optimize its parameters and check if it can obtain good metrics on the training data. If the model is unable to obtain satisfactory training metrics, it may be necessary to add more layers (i.e., increase its complexity).

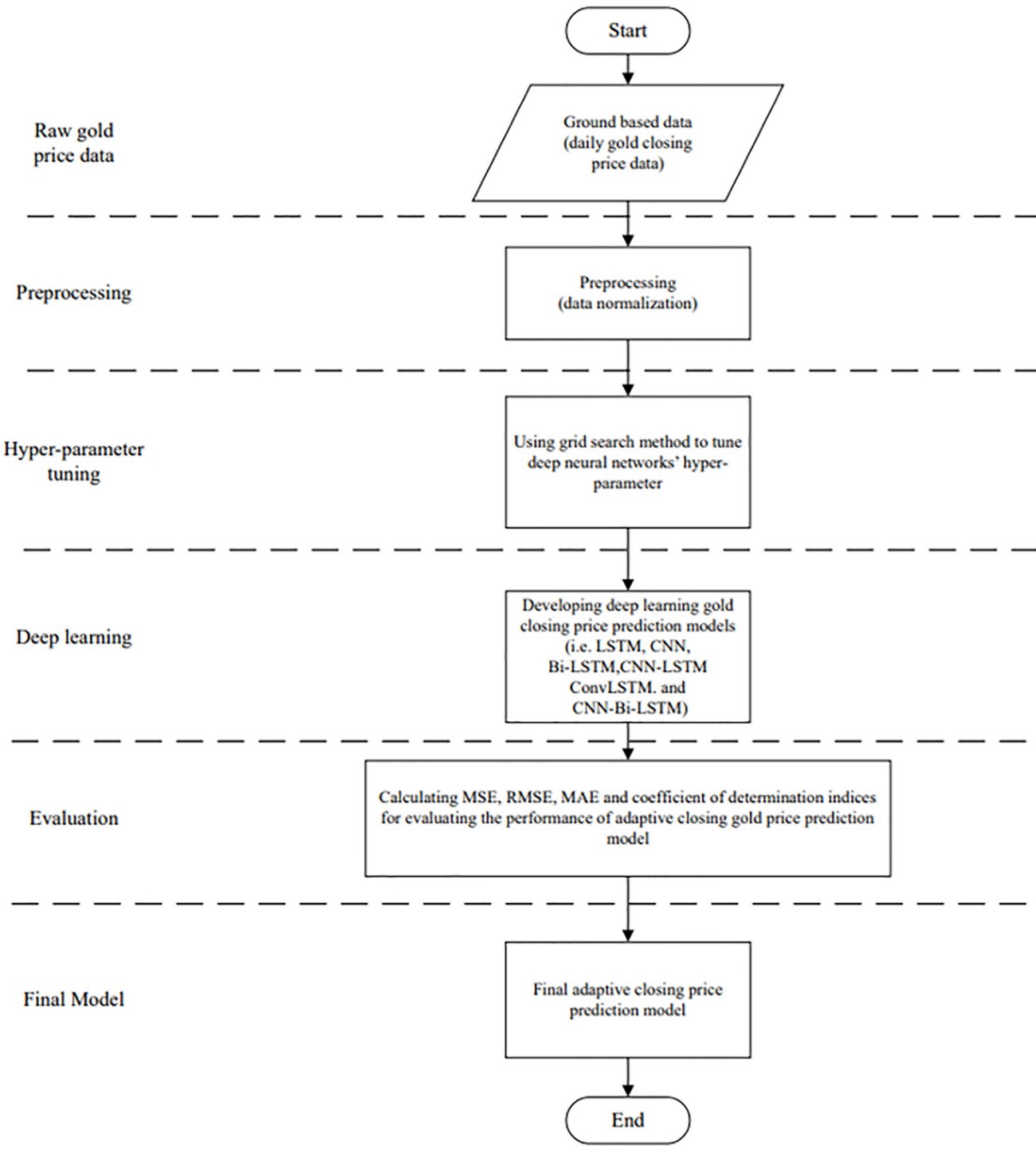

**Fig 3. Flowchart of the adaptive forecasting of the closing gold price.**

In comparison to the trial-and-error approach, tuning time for parameters is significantly reduced. Fig 5 illustrates the reference closing gold price and the predicted price in the test period using five tuned DNNs. As can be perceived from Fig 5, all models could predict the gold price movement well for test data. CNN-Bi-LSTM outperforms other models in total bias

**Table 2. The parameters' optimal value of DNNs (daily).**

|  | Stacked-LSTM | CNN | CNN-LSTM | ConvLSTM | Proposed CNN-Bi-LSTM |
|---|---|---|---|---|---|
| Lag time | 10 | 10 | 12 | 12 | 24 |
| # of sequences | - | - | 2 | 3 | 3 |
| # of hidden units | 40 | 20 | 10 | - | 20 |
| Filter size | - | 128 | 64 | 128 | 32 |
| Kernel size | - | 1 | 2 | (1,2) | 1 |
| Mini batch size | 512 | 256 | 1024 | 128 | 30 |
| Learning rate | 0.005 | 0.0005 | 0.005 | 0.005 | 0.0005 |
| Dropout probability | - | - | - | 0.2 | 0.1 |
| RMSE | 35.24 | 35.43 | 36.85 | 38.01 | 34.87 |
| RMAE | 4.96 | 5.19 | 5.24 | 6.65 | 5.15 |
| $R^2$ | 0.94 | 0.94 | 0.94 | 0.91 | 0.95 |

and captures extreme values, which are recommended buy and sell points. Furthermore, the proposed CNN-Bi-LSTM and CNN models adapt to trends and peaks faster than other models. In order to make better predictions, the trend of the closing gold price must quickly adapt to new prices. The rest of the models do not have the ability to use new data as much as the CNN-Bi-LSTM model. For instance, the CNN model has a problem in predicting buy points, low extremes, which come after sell points, including high prices in the period.

For the aforementioned learning rate, Fig 6 illustrates the variation of training and cross-validation losses versus epoch for the CNN-Bi-LSTM model. The loss (RMSE) value converges to a global minimum smoothly as it decreases. Furthermore, the difference between the losses of the cross-validation and training datasets is very small, which indicates that this model

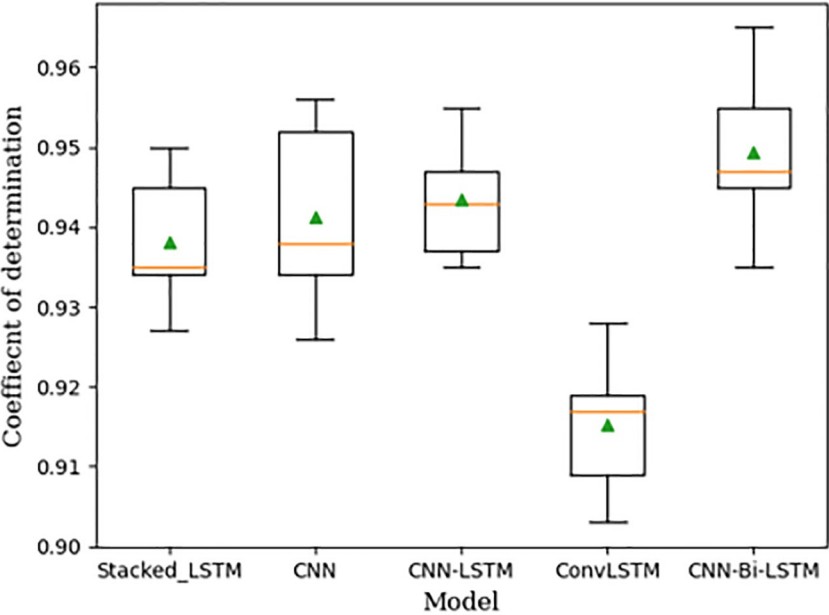

**Fig 4. Box plots of DNNs performance.** Each box plot is calculated from five realizations. The average coefficient of determination for the gold price forecasting in the validation set for (a) Stacked-LSTM, (b) CNN, (c) Bi-LSTM, (d) CNN-LSTM, (e) ConvLSTM, and (f) proposed CNN-Bi-LSTM (The green triangle and orange line indicate the mean and median coefficients of determination for each model realization).

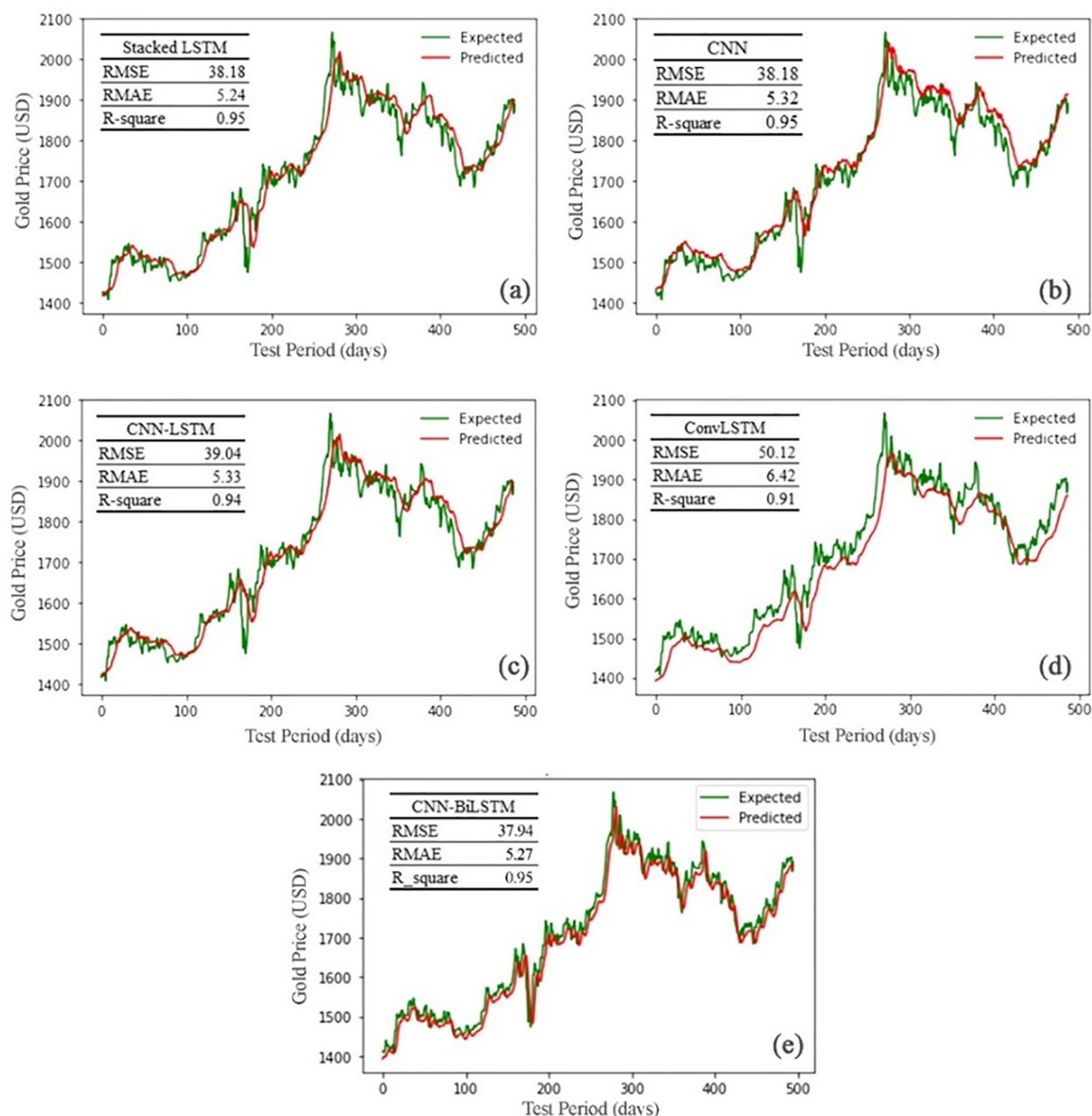

**Fig 5. Observation and forecasted charts for test data of (a) Stacked-LSTM, (b) CNN, (c) Bi-LSTM, (d) CNN-LSTM, (e) ConvLSTM, and (f) proposed CNN-Bi-LSTM.**

avoids overfitting. It has been demonstrated that the model converges well and remains stable after eight epochs.

The primary constraint of the aforementioned models is its reliance on a substantial volume of data in order to make accurate predictions.

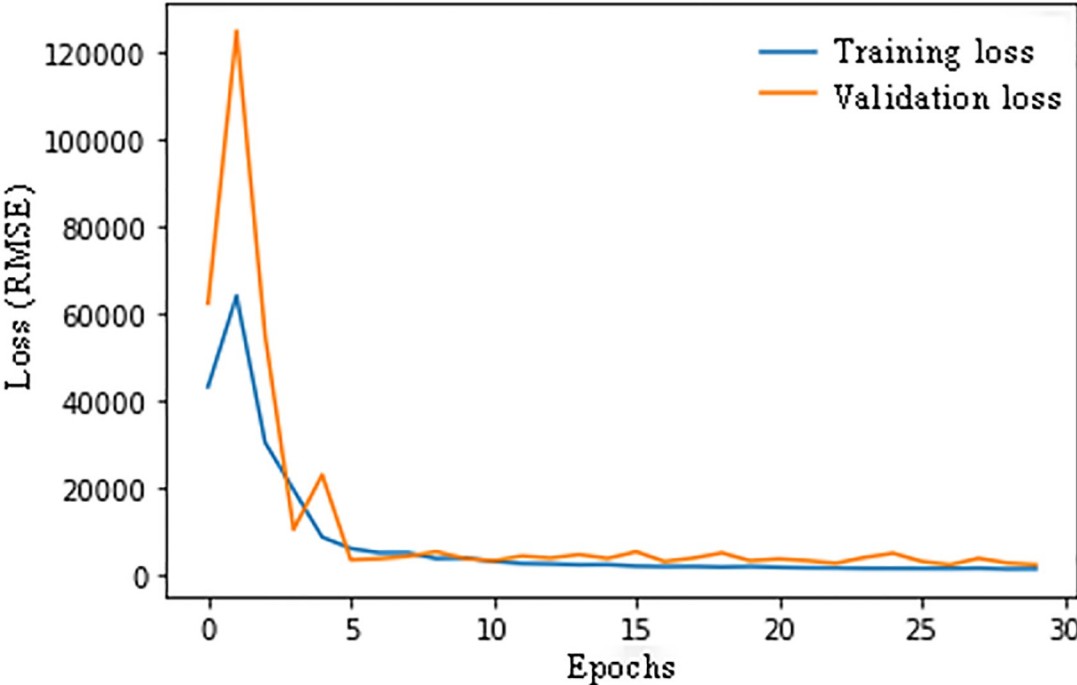

**Fig 6. Loss value versus epoch in CNN-Bi-LSTM.**

## 5.2 sensitivity analysis

In all mathematical modeling, sensitivity analysis should be an integral part of the process. An important benefit of performing a sensitivity analysis is identifying parameters or processes that are sensitive to changes in model output [53]. The sensitivity analysis of DNN modeling, like that of any mathematically based model, helps determine which input parameters have the greatest impact on the model's performance. The sensitivity analysis of the top three important hyperparameters of the CNN-Bi-LSTM (look back, learning rate, and mini-batch size) is shown in Fig 7.

**5.2.1 Lookback.** The lookback for the closing gold price was tested from 1 to 40-time steps. The results (Fig 7a) indicate that the CNN-Bi-LSTM network with a 24 look back has the highest average coefficient of determination, equal to 0.92. The accuracy of models with less look back than the optimal value is lower due to a lack of extreme data in the previous time step. Higher look back yields lower accuracy, which may be caused by the noise created by irrelevant input and the model's inability to predict extreme values that cause significant errors. In other words, models with more look back adapt more slowly and show a less wavy and smoother curve when extreme prices are involved in training sets. Based on the results, the observation of the last 24-look back is utilized in the rest of the analysis. For the Stacked-LSTM, CNN, CNN-LSTM, ConvLSTM, and proposed CNN-Bi-LSTM models, the gold price reference dataset from the last 1 to 40-time steps were tested, and the 10, 10, 12, 12, and 24-look back were set as optimal look backs in the DNNs.

**5.2.2 Learning rate.** Four different learning rates, 0.05, 0.005, 0.0005, and 0.00005, were applied for the proposed CNN-Bi-LSTM network. Results (Fig 7b) of five realizations indicate

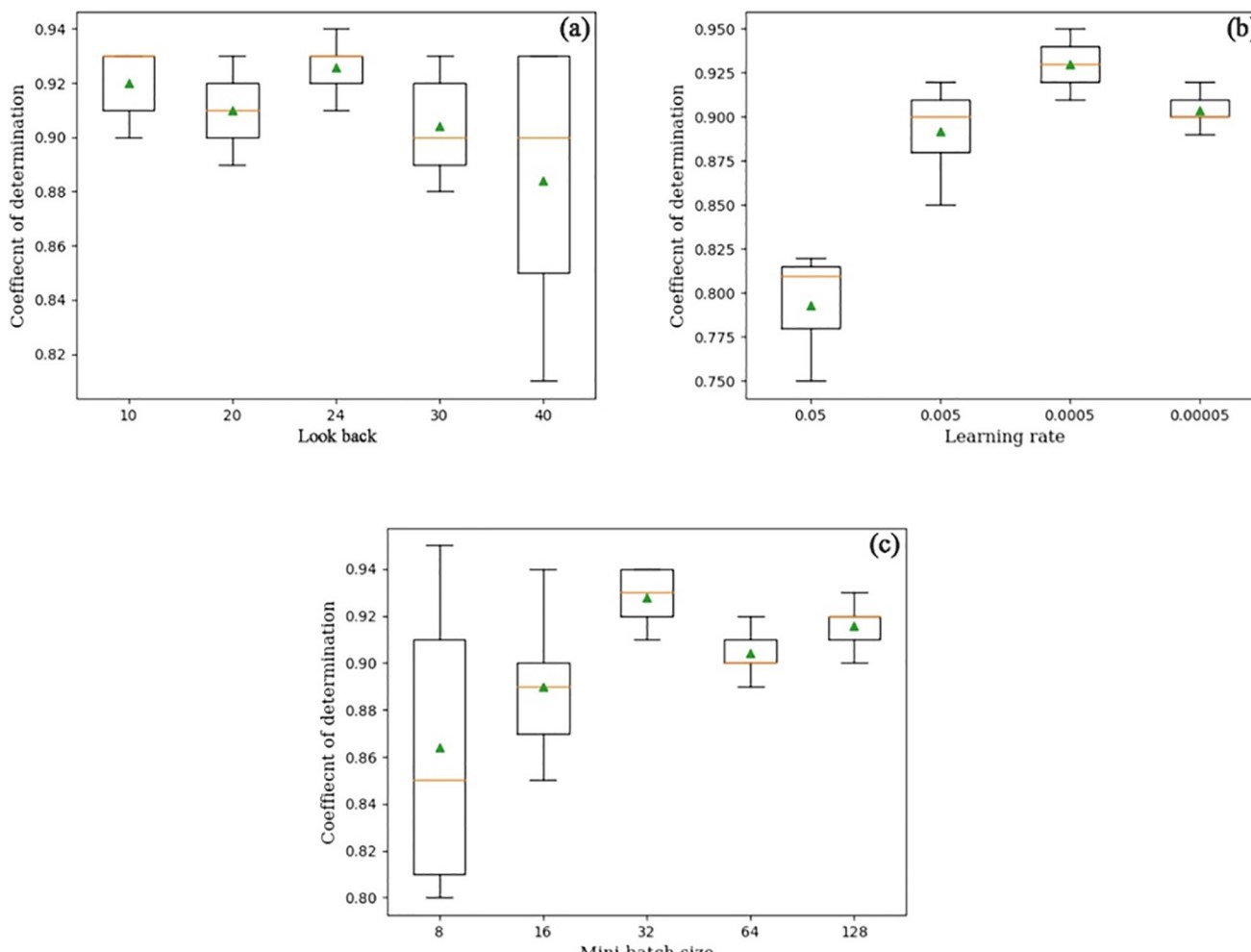

**Fig 7. Box plots of CNN-Bi-LSTM model.** Each box plot is calculated from five realizations. The average Coefficient of determination for the forecasting of the validation data for (a) look back, (b) learning rate, and (c) mini-batch sizes.

that the model with a 0.0005 learning rate has the highest coefficient of determination, possibly the most converging learning rate compared to other choices. It is noteworthy that the learning rate equal to 0.05 results in negative values in many runs, indicating a non-converging value.

**5.2.3 Mini-batch size.** In this analysis, the mini-batch size was adjusted from 8 to 128. Results (Fig 7c) illustrated that the average $R^2$ for the 32-mini-batch size fluctuates from 0.91 to 0.94. There is no significant difference in performance when using both 64 and 128 mini-batch sizes. Therefore, there could be various data in each mini-batch of the training dataset, which is necessary to calculate gradient descent in mini-batch sizes. Because there are insufficient extreme data in each mini-batch, the results lead to an unstable output.

TensorFlow was used as the backend to train all DNNs using a Python package called Keras [54]. For running the models, a computer system with GeForce GTX 1060 GPU and Intel(R) Core (TM) i7-7700HQ CPU is used.

## 6. Conclusion

This paper utilizes parameter tuning prior to the DNNs learning process for closing gold price forecasting on a daily time step. The closing gold price for 44 years, from 1978 to 2021, is

considered the input of models. The grid search tuning approach is applied to obtain the optimal performance of each DNN model. The forecasted closing gold price of LSTM, CNN, CNN-LSTM, ConvLSTM, and CNN-Bi-LSTM models are compared. The proposed CNN-Bi-LSTM model outperforms other models in general and can capture extreme values, which can be used as indications of where to buy or sell. Moreover, the CNN-Bi-LSTM model adapts to new closing gold price data more quickly than other models.

The performance of the proposed model depends on a number of characteristics. First, Due to a lack of extreme data in the previous time step, models with less look back than optimal value have lower accuracy. Lookback results in lower accuracy, which may be due to noise created by irrelevant inputs and the model's inability to predict extreme values. Second, according to the tuning procedure for the proposed CNN-Bi-LSTM model, the 32-mini batch size is considered a reliable value to reduce bias. Due to fluctuation in extreme data within each mini-batch, small mini-batch sizes result in unstable outputs. Compared to other models, the proposed CNN-Bi-LSTM performs better in total bias and capture of extreme values with coefficients of determination, RMSE, and RMAE equal to 0.95, 37.94, and 5.27, respectively, for the test data. Compared to other existing research, the following features distinguish the proposed framework:

- In order to predict closing gold prices, a novel model is implemented by combining CNN and LSTM features.

- In order to determine the optimal set of parameters for the CNN-Bi-LSTM model, the grid search tuning technique is used

- This paper performs a sensitivity analysis of three important parameters so as to evaluate the grid search tuning technique and identify parameters that are sensitive to changes in model output.

The closing gold price dataset is the only input sequence used by the suggested model. The model's performance may be enhanced by incorporating additional variables, such as the price of silver, crude oil, and the US dollar index, or by switching from grid search to other automatic hyperparameter tuning techniques like random search or Bayesian methods. Furthermore, the accuracy of the models may be improved by applying methods like wavelet transform, which reduce noise and use denoised data as input for deep learning models.

## Supporting information

**S1 Dataset.**
(ZIP)

## Author Contributions

**Writing – original draft:** Amirhossein Amini.

**Writing – review & editing:** Robab Kalantari.

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
