## [Decision Letter · Decision Letter 0]

6 Feb 2023

PONE-D-22-31551Gold Price Prediction by a CNN-Bi-LSTM model along with Automatic Parameter TuningPLOS ONE

Dear Dr. Kalantari,

Thank you for submitting your manuscript to PLOS ONE. After careful consideration, we feel that it has merit but does not fully meet PLOS ONE’s publication criteria as it currently stands. Therefore, we invite you to submit a revised version of the manuscript that addresses the points raised during the review process.

We look forward to receiving your revised manuscript.

Kind regards,

M. Shamim Kaiser, PhD

Academic Editor

PLOS ONE

Journal Requirements:

2. Please note that PLOS ONE has specific guidelines on code sharing for submissions in which author-generated code underpins the findings in the manuscript. In these cases, all author-generated code must be made available without restrictions upon publication of the work. Please review our guidelines at https://journals.plos.org/plosone/s/materials-and-software-sharing#loc-sharing-code and ensure that your code is shared in a way that follows best practice and facilitates reproducibility and reuse. New software must comply with the Open Source Definition.

Additional Editor Comments:

1. The literature review seems poor to me, authors must include recently published articles.

2. Add detail about the features.

3. Distinguish your work with the state of the art

Reviewers' comments:

Reviewer's Responses to Questions

**Comments to the Author**

1. Is the manuscript technically sound, and do the data support the conclusions?

Reviewer #1: Partly

Reviewer #2: Partly

2. Has the statistical analysis been performed appropriately and rigorously? 

Reviewer #1: Yes

Reviewer #2: Yes

3. Have the authors made all data underlying the findings in their manuscript fully available?

Reviewer #1: Yes

Reviewer #2: Yes

4. Is the manuscript presented in an intelligible fashion and written in standard English?

Reviewer #1: No

Reviewer #2: Yes

5. Review Comments to the Author

Reviewer #1: Although the article has corrected some errors in presentation, I keep my opinion as the previous review.

The article's contribution is not convincing enough for publication in PLOS ONE.

I suggest the author to submit the article in other more suitable journals about artificial intelligence

Reviewer #2: Thanks to authors for their contributions.

1. In the introduction: It is clearly stated the gap of research.

2. However, it will be better if they use different data-set source.

3. In conclusion, authors mentioned certain distinguish features of the proposed model in compare with existing models. -- I think it is better it should be write more precisely into the result-analysis section using certain statistical information.

6. PLOS authors have the option to publish the peer review history of their article (what does this mean?). If published, this will include your full peer review and any attached files.

Reviewer #1: No

Reviewer #2: No

---

## [Author Response · Author response to Decision Letter 0]

8 Mar 2023

Manuscript No. PONEDEC3

The authors would like to thank you for your kind reply to our submitted manuscript. We prepared a revised version of our manuscript with the title “Gold Price Prediction by a CNN-Bi-LSTM model along with

Automatic Parameter Tuning.” upon the useful comments of respected reviewers.

Response to reviewers' comments

At this time, please address the following queries: a) Please clarify the sources of funding (financial or material support) for your study. List the grants or organizations that supported your study, including funding received from your institution. 

b) State what role the funders took in the study. If the funders had no role in your study, please state: “The funders had no role in study design, data collection and analysis, decision to publish, or preparation of the manuscript.” c) If any authors received a salary from any of your funders, please state which authors and which funders. d) If you did not receive any funding for this study, please state: “The authors received no specific funding for this work.” The authors would like to thank the reviewer for the careful review of the manuscript, and for the comments, corrections, and suggestions. The last section is revised as recommended (Funding part) (L415).

Please include your amended statements within your cover letter; we will change the online submission form on your behalf. 4. In your Data Availability statement, you have not specified where the minimal data set underlying the results described in your manuscript can be found. PLOS defines a study's minimal data set as the underlying data used to reach the conclusions drawn in the manuscript and any additional data required to replicate the reported study findings in their entirety. All PLOS journals require that the minimal data set be made fully available. For more information about our data policy, please see http://journals.plos.org/plosone/s/data-availability.

Done. The gold price dataset is available at https://github.com/amirhosseinse7/Closing-gold-price. It is mentioned at L 416. 

Done! We added data as information files and public repository as aforementioned stated. Important: If there are ethical or legal restrictions to sharing your data publicly, please explain these restrictions in detail. Please see our guidelines for more information on what we consider unacceptable restrictions to publicly sharing data: http://journals.plos.org/plosone/s/data-availability#loc-unacceptable-data-access-restrictions. Note that it is not acceptable for the authors to be the sole named individuals responsible for ensuring data access. We will update your Data Availability statement to reflect the information you provide in your cover letter. Additional Editor Comments: 1. The literature review seems poor to me, authors must include recently published articles. Done! the manuscript has revised as recommended (L130-134) (Two recently published articles have added to this article)

2. Add detail about the features. The manuscript has revised as recommended (L176-177). About the features which extracted by CNN using time series there are not feature mapping method as there is in image sequence forecasting, so we just add the overall statement about the feature extraction of CNNs.

3. Distinguish your work with the state of the art The manuscript has revised as recommended (L72-80) and (L135-140).

Reviewers' comments: Reviewer's Responses to Questions Comments to the Author 1. Is the manuscript technically sound, and do the data support the conclusions? The manuscript must describe a technically sound piece of scientific research with data that supports the conclusions. Experiments must have been conducted rigorously, with appropriate controls, replication, and sample sizes. The conclusions must be drawn appropriately based on the data presented. Reviewer #1: Partly Reviewer #2: Partly

The manuscript (especially the discussion and conclusion) is revised as recommended. Thank you! 2. Has the statistical analysis been performed appropriately and rigorously? Reviewer #1: Yes Reviewer #2: Yes 3. Have the authors made all data underlying the findings in their manuscript fully available? The PLOS Data policy requires authors to make all data underlying the findings described in their manuscript fully available without restriction, with rare exception (please refer to the Data Availability Statement in the manuscript PDF file). The data should be provided as part of the manuscript or its supporting information, or deposited to a public repository. For example, in addition to summary statistics, the data points behind means, medians and variance measures should be available. If there are restrictions on publicly sharing data—e.g. participant privacy or use of data from a third party—those must be specified. Reviewer #1: Yes Reviewer #2: Yes 4. Is the manuscript presented in an intelligible fashion and written in standard English? PLOS ONE does not copyedit accepted manuscripts, so the language in submitted articles must be clear, correct, and unambiguous. Any typographical or grammatical errors should be corrected at revision, so please note any specific errors here. Reviewer #1: No Reviewer #2: Yes

Done. The text has been revised as suggested (changes are available through track change) 5. Review Comments to the Author Please use the space provided to explain your answers to the questions above. You may also include additional comments for the author, including concerns about dual publication, research ethics, or publication ethics. (Please upload your review as an attachment if it exceeds 20,000 characters) Reviewer #1: Although the article has corrected some errors in presentation, I keep my opinion as the previous review. The article's contribution is not convincing enough for publication in PLOS ONE. I suggest the author to submit the article in other more suitable journals about artificial intelligence. The article has changed in different sections. I hope this time the article’s contribution is convincing enough for publication in PLOS ONE. Reviewer #2: Thanks to authors for their contributions. 1. In the introduction: It is clearly stated the gap of research. 2. However, it will be better if they use different data-set source. Thank you for your suggestion. We utilize different datasets in our future research. 

3. In conclusion, authors mentioned certain distinguish features of the proposed model in compare with existing models. -- I think it is better it should be write more precisely into the result-analysis section using certain statistical information. Done! The manuscript has been revised as suggested. Thank you!

With the best regards

---

## [Decision Letter · Decision Letter 1]

6 Apr 2023

PONE-D-22-31551R1Gold Price Prediction by a CNN-Bi-LSTM model along with Automatic Parameter TuningPLOS ONE

Dear Dr. Kalantari,

Thank you for submitting your manuscript to PLOS ONE. After careful consideration, we feel that it has merit but does not fully meet PLOS ONE’s publication criteria as it currently stands. Therefore, we invite you to submit a revised version of the manuscript that addresses the points raised during the review process.

 It is advised that authors must complete the following: The literature review section must be updated. Addressed all the comments of Reviewer 1

We look forward to receiving your revised manuscript.

Kind regards,

M. Shamim Kaiser, PhD

Academic Editor

PLOS ONE

Reviewers' comments:

Reviewer's Responses to Questions

**Comments to the Author**

1. If the authors have adequately addressed your comments raised in a previous round of review and you feel that this manuscript is now acceptable for publication, you may indicate that here to bypass the “Comments to the Author” section, enter your conflict of interest statement in the “Confidential to Editor” section, and submit your "Accept" recommendation.

Reviewer #1: (No Response)

Reviewer #3: All comments have been addressed

2. Is the manuscript technically sound, and do the data support the conclusions?

Reviewer #1: Partly

Reviewer #3: Yes

3. Has the statistical analysis been performed appropriately and rigorously? 

Reviewer #1: Yes

Reviewer #3: Yes

4. Have the authors made all data underlying the findings in their manuscript fully available?

Reviewer #1: Yes

Reviewer #3: Yes

5. Is the manuscript presented in an intelligible fashion and written in standard English?

Reviewer #1: No

Reviewer #3: Yes

6. Review Comments to the Author

Reviewer #1: I still feel that the contribution of the paper is not enough for publication in PLOS ONE.

My suggestions are as follows:

1. Conduct survey of commodity price prediction studies such as:

+ Chen, Yufeng, Jinwang Wu, and Zhongrui Wu. "China’s commercial bank stock price prediction using a novel K-means-LSTM hybrid approach." Expert Systems with Applications 202 (2022): 117370.

+ Liu, Tengteng, et al. "A stock price prediction method based on meta-learning and variational mode decomposition." Knowledge-Based Systems 252 (2022): 109324.

+ Liu, Chong, et al. "An optimized nonlinear grey Bernoulli prediction model and its application in natural gas production." Expert Systems with Applications 194 (2022): 116448.

+ Vo, Anh H., Trang Nguyen, and Tuong Le. "Brent Oil Price Prediction Using Bi-LSTM Network." Intelligent Automation & Soft Computing 26.6 (2020).

+ Pindoriya, N. M., S. N. Singh, and S. K. Singh. "An adaptive wavelet neural network-based energy price forecasting in electricity markets." IEEE Transactions On power systems 23.3 (2008): 1423-1432.

…

2. Conduct the comparison with state-of-the-art methods for Gold Price Prediction such as:

+ Madziwa, Lawrence, Mallikarjun Pillalamarry, and Snehamoy Chatterjee. "Gold price forecasting using multivariate stochastic model." Resources Policy 76 (2022): 102544.

+ Lu, Wanbo, et al. "International Gold Price Forecast Based on CEEMDAN and Support Vector Regression with Grey Wolf Algorithm." Complexity 2022 (2022).

+ Liang, Yanhui, Yu Lin, and Qin Lu. "Forecasting gold price using a novel hybrid model with ICEEMDAN and LSTM-CNN-CBAM." Expert Systems with Applications 206 (2022): 117847.

…

Reviewer #3: (No Response)

7. PLOS authors have the option to publish the peer review history of their article (what does this mean?). If published, this will include your full peer review and any attached files.

Reviewer #1: No

Reviewer #3: **Yes: **Sobhana Jahan

---

## [Author Response · Author response to Decision Letter 1]

10 May 2023

Response to reviewers' comments (Revision 2)

The authors would like to thank the reviewer for the careful review of the manuscript, and for the comments, corrections, and suggestions. This document provides a reply to the comments of reviewer. In order to respond, we have added some paragraphs and explained in different sections

General comment

It is advised that authors must complete the following: 

• The literature review section must be updated. Done! The recent studies section has been changed as suggested. Thank you for your constructive comment!

• Addressed all the comments of Reviewer 1. The manuscript has been revised as suggested. 

Reviewer #1 comments:

I still feel that the contribution of the paper is not enough for publication in PLOS ONE.

My suggestions are as follows:

1. Conduct survey of commodity price prediction studies such as:

+ Chen, Yufeng, Jinwang Wu, and Zhongrui Wu. "China’s commercial bank stock price prediction using a novel K-means-LSTM hybrid approach." Expert Systems with Applications 202 (2022): 117370.

+ Liu, Tengteng, et al. "A stock price prediction method based on meta-learning and variational mode decomposition." Knowledge-Based Systems 252 (2022): 109324.

+ Liu, Chong, et al. "An optimized nonlinear grey Bernoulli prediction model and its application in natural gas production." Expert Systems with Applications 194 (2022): 116448.

+ Vo, Anh H., Trang Nguyen, and Tuong Le. "Brent Oil Price Prediction Using Bi-LSTM Network." Intelligent Automation & Soft Computing 26.6 (2020).

+ Pindoriya, N. M., S. N. Singh, and S. K. Singh. "An adaptive wavelet neural network-based energy price forecasting in electricity markets." IEEE Transactions On power systems 23.3 (2008): 1423-1432.

2. Conduct the comparison with state-of-the-art methods for Gold Price Prediction such as:

+ Madziwa, Lawrence, Mallikarjun Pillalamarry, and Snehamoy Chatterjee. "Gold price forecasting using multivariate stochastic model." Resources Policy 76 (2022): 102544.

This research is related to prediction of annual gold price which is less volatile compared to daily gold price. In addition, the researchers used traditional models like AR based model

+ Lu, Wanbo, et al. "International Gold Price Forecast Based on CEEMDAN and Support Vector Regression with Grey Wolf Algorithm." Complexity 2022 (2022).

+ Liang, Yanhui, Yu Lin, and Qin Lu. "Forecasting gold price using a novel hybrid model with ICEEMDAN and LSTM-CNN-CBAM." Expert Systems with Applications 206 (2022): 117847.

Thank you for your constructive comment. Done! The aforementioned studies are added to the manuscript as suggested. 

1) Other methods mentioned in part one will be explored in our second article about gold price forecasting using the combination of tuned DNNs, various decomposition algorithms, and ensemble models.

2) Regarding the comparison part, due to the different lengths of test data and overall data, the results of different studies are not quite comparable. For example, in Lu et al. (2022) the researchers just utilized the last 5% of data (in our case last 10%) as test data and the length of the data was different from ours.

---

## [Decision Letter · Decision Letter 2]

15 Aug 2023

PONE-D-22-31551R2Gold Price Prediction by a CNN-Bi-LSTM model along with Automatic Parameter TuningPLOS ONE

Dear Dr. Kalantari,

Thank you for submitting your manuscript to PLOS ONE. After careful consideration, we feel that it has merit but does not fully meet PLOS ONE’s publication criteria as it currently stands. Therefore, we invite you to submit a revised version of the manuscript that addresses the points raised during the review process.

All comments suggested by the reviewers must be addressed. The quality of the images are poor. Please improve it. Include recently published articles

We look forward to receiving your revised manuscript.

Kind regards,

M. Shamim Kaiser, PhD

Academic Editor

PLOS ONE

Journal Requirements:

Reviewers' comments:

Reviewer's Responses to Questions

**Comments to the Author**

1. If the authors have adequately addressed your comments raised in a previous round of review and you feel that this manuscript is now acceptable for publication, you may indicate that here to bypass the “Comments to the Author” section, enter your conflict of interest statement in the “Confidential to Editor” section, and submit your "Accept" recommendation.

Reviewer #1: (No Response)

Reviewer #2: (No Response)

Reviewer #4: (No Response)

2. Is the manuscript technically sound, and do the data support the conclusions?

Reviewer #1: Partly

Reviewer #2: Yes

Reviewer #4: Yes

3. Has the statistical analysis been performed appropriately and rigorously? 

Reviewer #1: No

Reviewer #2: Yes

Reviewer #4: Yes

4. Have the authors made all data underlying the findings in their manuscript fully available?

Reviewer #1: No

Reviewer #2: Yes

Reviewer #4: Yes

5. Is the manuscript presented in an intelligible fashion and written in standard English?

Reviewer #1: No

Reviewer #2: Yes

Reviewer #4: Yes

6. Review Comments to the Author

Reviewer #1: I am not satisfied with the author's answer on the comparison part.

Besides, I found the topic of the article to be trivial, nothing new, the scientific quality of the article was not satisfactory for publication in PLOS ONE.

Therefore I suggest reject.

Reviewer #2: Certain observations:

1. Chen et al. [24] - you mentioned "more accurate than other methods".

Its better to mention any metrics that represent the mentioned phrase.

2. Liu et al. [25] - similar explanation like 1.

3. Liang et al. [27] -- its better to mention the metrics that represent the term "more accurate".

The best solution, I think, make a table and mention the result and properties of the mentioned schemes and others.

Reviewer #4: The paper presents a novel CNN-Bi-LSTM model for closing gold price forecasting. The model is well-designed and the experimental results are convincing. However, there are a few minor concerns that need to be addressed before the paper can be accepted for publication.

1. The paper only uses the closing gold price dataset. It would be interesting to see how the model performs when other influential factors, such as silver price, crude oil, and the U.S. dollar index, are included in the dataset.

2. The author could also provide more discussion of the limitations of the model.

3. The author mentions that data augmentation methods, such as adding noise to the data, may improve the performance of deep neural networks. However, they do not discuss how they might implement these methods in their models or how these augmentation methods might affect the accuracy of the predictions.

4. The paper only uses grid search for hyperparameter tuning. It would be beneficial to compare the results of grid search with other hyperparameter tuning methods, such as random search and Bayesian optimization.

Overall, the paper is well-written and the research is sound. However, the minor concerns mentioned above need to be addressed before the paper can be accepted for publication.

7. PLOS authors have the option to publish the peer review history of their article (what does this mean?). If published, this will include your full peer review and any attached files.

Reviewer #1: No

Reviewer #2: No

Reviewer #4: No

---

## [Author Response · Author response to Decision Letter 2]

12 Oct 2023

Dear Editors of PlOS ONE

The authors would like to thank for your kind reply on our submitted manuscript. We prepared revised version of our manuscript (No. PONE-D-22-31551R2) upon the useful comments of respected reviewers. Before listing response to the comment.

Response to reviewers' comments (Revision 3)

All comments suggested by the reviewers must be addressed. The quality of the images is poor. Please improve it. 

Done! The quality of all Figures is improved (Manuscript file)

Include recently published articles

Done! the manuscript has revised as (L162-170) (considering Reviewer #2 comments in these studies)

Reviewers' comments: Comments to the Author 1. If the authors have adequately addressed your comments raised in a previous round of review and you feel that this manuscript is now acceptable for publication, you may indicate that here to bypass the “Comments to the Author” section, enter your conflict-of-interest statement in the “Confidential to Editor” section, and submit your "Accept" recommendation. Reviewer #1: (No Response) Reviewer #2: (No Response) Reviewer #4: (No Response) 

2. Is the manuscript technically sound, and do the data support the conclusions? The manuscript must describe a technically sound piece of scientific research with data that supports the conclusions. Experiments must have been conducted rigorously, with appropriate controls, replication, and sample sizes. The conclusions must be drawn appropriately based on the data presented. Reviewer #1: Partly Reviewer #2: Yes Reviewer #4: Yes 

3. Has the statistical analysis been performed appropriately and rigorously? Reviewer #1: No Reviewer #2: Yes Reviewer #4: Yes 

4. Have the authors made all data underlying the findings in their manuscript fully available? The PLOS Data policy requires authors to make all data underlying the findings described in their manuscript fully available without restriction, with rare exception (please refer to the Data Availability Statement in the manuscript PDF file). The data should be provided as part of the manuscript or its supporting information, or deposited to a public repository. For example, in addition to summary statistics, the data points behind means, medians and variance measures should be available. If there are restrictions on publicly sharing data—e.g. participant privacy or use of data from a third party—those must be specified. Reviewer #1: No The gold price dataset is available at https://github.com/amirhosseinse7/Closing-gold-price. It is mentioned in L 416. 

Reviewer #2: Yes Reviewer #4: Yes 

5. Is the manuscript presented in an intelligible fashion and written in standard English? PLOS ONE does not copyedit accepted manuscripts, so the language in submitted articles must be clear, correct, and unambiguous. Any typographical or grammatical errors should be corrected at revision, so please note any specific errors here. Reviewer #1: No Done. The text has been revised as suggested.

Reviewer #2: Yes Reviewer #4: Yes 

6. Review Comments to the Author Please use the space provided to explain your answers to the questions above. You may also include additional comments for the author, including concerns about dual publication, research ethics, or publication ethics. (Please upload your review as an attachment if it exceeds 20,000 characters) Reviewer #1: I am not satisfied with the author's answer on the comparison part. Besides, I found the topic of the article to be trivial, nothing new, the scientific quality of the article was not satisfactory for publication in PLOS ONE. Therefore, I suggest reject. We have evaluated the Light Gradient Boosted Machine (LightGBM) ensemble model but were unable to improve statistical criteria; therefore, we have omitted this model from our article.

Reviewer #2: Certain observations: 1. Chen et al. [24] - you mentioned "more accurate than other methods". It’s better to mention any metrics that represent the mentioned phrase. Done. The manuscript has been changed as suggested (L142-144). Thank you!

2. Liu et al. [25] - similar explanation like 1.

The text has been modified as suggested (L150)

3. Liang et al. [27] -- its better to mention the metrics that represent the term "more accurate". Done. The manuscript has been changed as suggested (L174-175).

The best solution, I think, make a table and mention the result and properties of the mentioned schemes and others. In order to ensure the comprehensiveness of this table, we think it is better to include other relevant publications that are referenced in the methodology and results sections. This inclusion will necessitate a revision of the overall structure of our article. However, in the event that it is deemed essential, we shall incorporate the aforementioned chart.

Reviewer #4: The paper presents a novel CNN-Bi-LSTM model for closing gold price forecasting. The model is well-designed and the experimental results are convincing. However, there are a few minor concerns that need to be addressed before the paper can be accepted for publication. 1. The paper only uses the closing gold price dataset. It would be interesting to see how the model performs when other influential factors, such as silver price, crude oil, and the U.S. dollar index, are included in the dataset. As we have previously addressed the multivariate predicting of closing gold price by incorporating several relevant elements in a separate article, we refrain from including these components in the present paper. This topic was addressed in the concluding section. Thank you for this comment!

2. The author could also provide more discussion of the limitations of the model. The primary constraint of the aforementioned model is its reliance on a substantial volume of data in order to make accurate predictions. This text has been added in L388.

3. The author mentions that data augmentation methods, such as adding noise to the data, may improve the performance of deep neural networks. However, they do not discuss how they might implement these methods in their models or how these augmentation methods might affect the accuracy of the predictions. The performance of models is contingent upon the quantity of data available and the ability to discern patterns within it. Consequently, augmenting the dataset through the utilization of deep learning models like Generative Adversarial Networks (GANs) or employing data augmentation techniques is expected to enhance the accuracy of the outcomes. Moreover, the application of techniques such as wavelet transform, which involves noise reduction and utilization of the denoised data as input for deep learning models, has the potential to enhance the precision of the outcomes. This description has added in (L459-465).

4. The paper only uses grid search for hyperparameter tuning. It would be beneficial to compare the results of grid search with other hyperparameter tuning methods, such as random search and Bayesian optimization. The rationale for not comparing alternative hyperparameter tuning approaches in our study was mostly due to the near-perfect performance achieved by the various models under consideration. Additionally, it was anticipated that these alternative methods would converge to the same hyperparameter values as those obtained by grid search.

Overall, the paper is well-written and the research is sound. However, the minor concerns mentioned above need to be addressed before the paper can be accepted for publication. Thank you!

7. PLOS authors have the option to publish the peer review history of their article (what does this mean?). If published, this will include your full peer review and any attached files. Do you want your identity to be public for this peer review? For information about this choice, including consent withdrawal, please see our Privacy Policy. Reviewer #1: No Reviewer #2: No Reviewer #4: No 

With the best regards

R. Kalantari

---

## [Decision Letter · Decision Letter 3]

14 Dec 2023

PONE-D-22-31551R3Gold Price Prediction by a CNN-Bi-LSTM model along with Automatic Parameter TuningPLOS ONE

Dear Dr. Kalantari,

Thank you for submitting your manuscript to PLOS ONE. After careful consideration, we feel that it has merit but does not fully meet PLOS ONE’s publication criteria as it currently stands. Therefore, we invite you to submit a revised version of the manuscript that addresses the points raised during the review process.

We look forward to receiving your revised manuscript.

Kind regards,

Jayesh Soni

Academic Editor

PLOS ONE

Journal Requirements:

Additional Editor Comments:

Your work presents an intriguing approach to predicting gold prices using deep learning techniques.

However, I have a few queries that I believe, once addressed, could further strengthen your manuscript:

Given the unpredictability of financial markets, particularly during extraordinary events like the 2008 financial crisis or the recent pandemic, how does your model perform under such atypical conditions? Is there evidence of its adaptability and robustness in these scenarios?

While your model shows significant promise, a comparative analysis with non-CNN and non-LSTM based machine learning models could offer a broader perspective on its relative performance. Have such comparisons been considered or conducted?

Looking forward, what are the avenues for enhancement in your model? Are there upcoming methodologies or data sources that could be leveraged to improve prediction accuracy?

Reviewers' comments:

Reviewer's Responses to Questions

**Comments to the Author**

1. If the authors have adequately addressed your comments raised in a previous round of review and you feel that this manuscript is now acceptable for publication, you may indicate that here to bypass the “Comments to the Author” section, enter your conflict of interest statement in the “Confidential to Editor” section, and submit your "Accept" recommendation.

Reviewer #3: All comments have been addressed

2. Is the manuscript technically sound, and do the data support the conclusions?

Reviewer #3: Yes

3. Has the statistical analysis been performed appropriately and rigorously? 

Reviewer #3: Yes

4. Have the authors made all data underlying the findings in their manuscript fully available?

Reviewer #3: Yes

5. Is the manuscript presented in an intelligible fashion and written in standard English?

Reviewer #3: Yes

6. Review Comments to the Author

Reviewer #3: (No Response)

7. PLOS authors have the option to publish the peer review history of their article (what does this mean?). If published, this will include your full peer review and any attached files.

Reviewer #3: No

---

## [Author Response · Author response to Decision Letter 3]

18 Dec 2023

Dear Jayesh Soni

Academic Editor of PlOS one

The authors would like to thank you for your kind reply to our submitted manuscript. We prepared a revised version of our manuscript No. PONE-D-22-31551R3 upon the useful comments of respected reviewers.

Manuscript No. PONE-D-22-31551R3

Response to reviewers' comments (Revision 3)

Additional Editor Comments: Your work presents an intriguing approach to predicting gold prices using deep learning techniques. However, I have a few queries that I believe, once addressed, could further strengthen your manuscript: Given the unpredictability of financial markets, particularly during extraordinary events like the 2008 financial crisis or the recent pandemic, how does your model perform under such atypical conditions? Is there evidence of its adaptability and robustness in these scenarios? In LSTM-based models, the number of lag times (time steps) determines how and when these models adjust to atypical conditions, such as the ones you described. As Fig. 5 illustrates, different DNN models quickly adjust to the sequence's peaks points (small scale atypical condition) (depending on the lag time, their reaction time changes). However, a detailed analysis of these models' accuracy in atypical scenarios (large scale atypical conditions) will be investigated in future studies.

While your model shows significant promise, a comparative analysis with non-CNN and non-LSTM based machine learning models could offer a broader perspective on its relative performance. Have such comparisons been considered or conducted? Due to the findings of earlier studies in literature review indicating that DNN-based models outperform ANN models across multiple domains, we do not perform comparative analysis with non-CNN and non-LSTM based machine learning models (such as MLP, RBF, GRNN, etc.) (L59-66)

Looking forward, what are the avenues for enhancement in your model? Are there upcoming methodologies or data sources that could be leveraged to improve prediction accuracy? Thank you, the text has been changed due to this productive question. The model's performance may be enhanced by incorporating additional variables, such as the price of silver, crude oil, and the US dollar index, or by switching from grid search to other automatic hyperparameter tuning techniques like random search or Bayesian methods. Furthermore, the accuracy of the models may be improved by applying methods like wavelet transform, which reduce noise and use denoised data as input for deep learning models (L456-461).

With the best regards

---

## [Decision Letter · Decision Letter 4]

25 Jan 2024

Gold Price Prediction by a CNN-Bi-LSTM model along with Automatic Parameter Tuning

PONE-D-22-31551R4

Dear Dr. Robab Kalantari,

We’re pleased to inform you that your manuscript has been judged scientifically suitable for publication and will be formally accepted for publication once it meets all outstanding technical requirements.

Kind regards,

Jayesh Soni

Academic Editor

PLOS ONE

Additional Editor Comments (optional):

All the comments are addressed.

Reviewers' comments:

Reviewer's Responses to Questions

**Comments to the Author**

6. Review Comments to the Author

Reviewer: satisfied with authors response. They have modified the paper in best mode. The paper is in the desired stage to be proceed to publish

---

## [Editor Report · Acceptance letter]

27 Feb 2024

PONE-D-22-31551R4 

PLOS ONE

Dear Dr. Kalantari, 

I'm pleased to inform you that your manuscript has been deemed suitable for publication in PLOS ONE. Congratulations! Your manuscript is now being handed over to our production team.

Kind regards, 

on behalf of

Dr. Jayesh Soni 

Academic Editor

PLOS ONE